# Factors of Food Waste Reduction Underlying the Extended Theory of Planned Behavior: A Study of Consumer Behavior towards the Intention to Reduce Food Waste

Johannes Schrank [1,2], Aphinya Hanchai [1,2], Sahapob Thongsalab [1,2], Narakorn Sawaddee [1,2], Kirana Chanrattanagorn [1,2] and Chavis Ketkaew [1,2,*]

1 International College, Khon Kaen University, Khon Kaen 40002, Thailand
2 Center for Sustainable Innovation and Society, Khon Kaen University, Khon Kaen 40002, Thailand
* Correspondence: chaket@kku.ac.th

**Abstract:** Food waste represents an economic, environmental, and social threat, which makes it an important subject of investigation. Food waste behavior has a crucial effect on everyone's food security, food safety, economic growth, and the environment; hence, it requires further analysis. The article's objective is to study the food waste reduction behavior of individual consumers and to examine factors which can explain the intention to reduce food waste. The study's conceptual foundation is the Theory of Planned Behavior (TPB), which aims to explain the relationship between an individual's attitudes, subjective norms, and perceived behavioral control. The paper extends the TPB by including new factors such as environmental concern, perceived ascription of responsibility, marketing addiction, moral norm, and waste preventing behavior. The data were collected via quota sampling and examined using the structural equation modeling (SEM). The study employed a sample of 369 people in Thailand. The results show that waste preventing behavior, attitude, and perceived behavioral control significantly impact the intention to reduce food waste. The subjective norm and environmental concern positively affects the attitude, which subsequently impacts the intention to reduce food waste. Marketing addiction negatively impacts perceived behavioral control and, hence, increases food waste. This research paper enlarges the understanding of the intention to minimize food waste. Moreover, it points out the implications on how consumers and the government may improve the desire to decrease food waste.

**Keywords:** attitude; consumer behavior; food waste; theory of planned behavior; waste reduction





## 1. Introduction

Food waste is a significant problem around the globe. Food waste reduction is a crucial issue that requires attention due to its negative impact on the environment, economy, and society. Around one-third of the food produced worldwide is never consumed, leading to significant greenhouse gas emissions, wasted resources, and increased hunger and poverty [1,2]. Food waste generates toxic gases and, therefore, presents a serious risk to human health and the environment [3]. In addition, food waste results in a loss of limited resources including land, water, and energy [4]. A study of food waste in the sector of tourism and food service found that wasted food primarily in restaurants and hotels can be attributed to various factors, such as overproduction, poor forecasting, and customer behavior [5]. Aschemann-Witzel et al. [6] found that factors such as income, price, and the appearance of food can influence consumer behavior and contribute to food waste. Dos Santos et al. [7] concluded that consumer behavior is a significant factor in the generation of food waste. A recent study by Principato et al. [8] summarized the various consumer-level aspects of the food waste phenomena and proposed frameworks to explain food waste behavior. Massive amounts of waste negatively affect the environment, which does not only impact Thailand but can also lead to global problems [9]. Food waste can also produce

large amounts of methane. Methane has a greater global warming potential than carbon dioxide [10]. According to Munesue et al. [11], food waste generates a significant amount of greenhouse gas emissions and contributes to environmental degradation. In addition to the inadequate management of the already-existing food waste, the primary issue with food waste is the unnecessarily high agricultural production.

The food waste hierarchy best describes the priorities when managing food waste. The most favorable option is prevention, followed by re-use, recycle, recovery, and disposal as the least favorable option [12]. It is the responsibility of households and individual consumers to contribute to the reduction of food waste by using the 3R principle which consists of Reduce, Reuse, and Recycle [13]. Using this 3R principle can reduce the use of resources as well. Reduce means to decrease the amount of food and to manage the amount of food, i.e., not hoarding too much food. Reuse is the use of leftover ingredients to cook a new menu and to keep leftover food that can still be eaten for the next meal, which will assist in minimizing the quantity of wasted food. Recycle is the use of food waste to make compost or animal feed. Following the 3R principles, consumers can help to reduce economic losses, reduce the use of natural resources, reduce global warming, and protect the environment. Consumer perceptions and behaviors are the key drivers in food waste reduction [14]. There are various factors of food waste reduction, including environmental awareness, economic incentives, and cultural norms [15]. To reduce the amount of food waste, it is crucial to address the root causes of the problem at all steps of the food supply chain, from farming to eating [16]. Food waste can be significantly decreased by taking simple steps like meal planning, creating shopping lists, utilizing leftovers, and properly storing food [17]. Governments and businesses can also take steps to reduce the quantity of food waste by implementing policies, e.g., food waste reduction targets and suppressing and recycling food loss [18]. According to our estimates, a third of the globally produced food for people's use is wasted [19,20]. The amount of wasted food is a problem in developed countries where it contributes significantly to household waste [21].

Food waste in Thailand is a growing issue, as the country's economic development has led to a rise in food production and consumption. According to a study, the amount of food waste produced in Thailand is estimated to be 9.3 million tons annually [22]. A study indicated that food waste generates the largest part of all waste in Bangkok between 42% and 45% [23]. This paper focuses on both edible and inedible food waste. Edible food waste refers to food that is still safe and suitable for human consumption but is wasted. Inedible food waste, on the other hand, refers to the parts of food that are not intended and are no longer safe for human consumption. In developing countries, the key sources of food waste are households, supermarkets, and the hospitality industry [24]. The lack of proper waste management infrastructure and education about the impacts of food waste pose significant challenges to minimize food waste in Thailand [25]. Reducing food waste in households, where it originates, is one strategy to reduce the quantity of waste. Thailand has taken various actions to achieve its goal of halving food waste within the next 10 years under the Sustainable Development Goals (SCGs). In order to meet the goals, the Pollution Control Department in Thailand is responsible for studying, analyzing, and formulating strategies, guidelines, and measures to minimize food waste, including tracking the amount of food waste and managing it effectively. The key factor of food waste reduction remains to be consumer behavior, which is studied in this paper.

This article aims to illustrate the awareness of food waste behavior and aspects for the reduction of food waste, as well as analyze the waste-preventing behavior of individuals and investigate the factors which can explain the intention to reduce food waste. This study focuses on the behavioral characteristics of consumers. In addition to examining how the difficulties of food waste reduction may be handled within the framework of a self-manageable basis, it seeks to identify the elements that contribute to food waste formation. This paper analyzes the factors which affects the intention to minimize food waste based on the theory of planned behavior (TPB), with the variables' subjective norm,

attitude, perceived behavioral control, and the intention to reduce food waste. Furthermore, it extends the TPB by adding other important factors (waste preventing behavior, perceived ascription of responsibility, moral norm, environmental concern, and marketing addiction) which impact the intention to reduce wasted food. The moral norm is an important factor which influences attitudes, which in turn may impact the intention. Thus, the moral norm might be a key effect on the reduction of wasted food and must be included in the research framework. A conceptual framework was created in earlier studies to analyze consumer behavior with reference to food waste. However, to further explore the intention and behavior towards the waste of food, additional studies which cover more factors are required. This paper fills that gap. The TPB is able to explain the individual's behavior. This study examines whether its main variables and the addition of the new variables have a substantial impact on the goal of reducing the waste of food. Moreover, it tries to find reasons and determinants for the especially high amounts of food waste in Thailand. It analyzes why Thai people generate significant amounts of food waste and how to decrease it. The existing literature studied factors which impact the intention to minimize wasted food by using the TPB, but important variables were omitted. This article fills the gap by expanding the TPB with relevant factors related to food waste, such as waste-preventing behavior. Hence, the paper shows a more complete picture of consumer behavior towards food waste.

This study is structured as follows: A literature review and the formulation of hypotheses begins in Section 2. Then, Section 3 outlines the research methodology, sample and data gathering, and development of measures. Section 4 shows the analyses and results. The study's theoretical and practical implications are discussed in Section 5. Section 6 of this research paper serves as a summary and discusses the major findings from this study.

## 2. Literature Review

Numerous studies related to food waste reduction applied the structural equation modeling (SEM) approach. Several studies found relationships between consumer attitudes, the subjective norm, perceived behavioral control, and behavioral intention [26–28]. Based on past studies, the relation of variables might be shaped using the SEM. The theories and literature that assisted in creating a model and hypotheses are discussed in detail in the following sections.

### 2.1. Theory of Planned Behavior

The theory of planned behavior (TPB) seeks to explain how a person's attitudes, subjective norms, and perception of behavioral control relate to their intention to engage in a certain behavior. The theory of planned behavior includes the subjective norm, attitude, and perceived behavior control. Dowd et al. [26] applied the TPB in a paper which explains consumers' food choices. The TPB has proven to be an effective hypothesis for anticipating and forecasting customer behavior. The TPB is beneficial in forecasting the consumers' intention to purchase food [29]. Wang and Wang [30] researched the TPB and found that the three most crucial components of green food are commitment, perceived behavioral control, and perceived knowledge. Wongsaichia et al. [27] identified the factors influencing the intention to purchase food based on the TPB. Heidari et al. [28] studied the food waste reduction behavior by applying the TPB. The TPB can be used to understand how individual attitudes, perceived social pressure, and perceived behavioral control influence the intention to minimize food waste. A study showed that the TPB can be applied to predict consumer behavior towards food [31]. The TPB has been applied successfully for understanding food waste prevention and reduction [32].

### 2.2. Perceived Ascription of Responsibility (PAR)

Global food waste is often caused by human eating behaviors both at home and outside of it, but food waste cannot be determined by a single behavior because it has to go through a combination of behaviors to achieve it [33]. Human food waste can affect

many aspects, such as the environment, the economy, or society. Almost half of the quantity of food waste is produced at the household level, making this level one of the greatest producers of wasted food [34]. PAR can affect the subjective norm positively [28]. Based on the existing literature, we can derive the following hypothesis.

**H1.** *PAR positively impacts the subjective norm towards reducing wasted food.*

### 2.3. Moral Norm

Humans need to focus on food waste for social responsibility, starting with the problem of food waste in the household. Consumers can be socially responsible by reducing their daily food waste. Mondéjar-Jiménez et al. [35] found that the moral norm has a large effect on people's desire to decrease food waste. Moreover, Heidari et al. [28] found that moral attitude has a significant effect on attitudes towards food waste. The moral norm may impact attitudes towards the reduction of wasted food and, therefore, is an important element of the research framework. This leads to the following hypothesis.

**H2.** *The moral norm of human behavior has a positive impact on the attitude towards decreasing food waste.*

### 2.4. Environmental Concern

If handled properly, the waste management sector and waste prevention can reduce global greenhouse gas emissions significantly. Lin and Guan [36] found that consumers who have a higher level of concern for the environment are more likely to adopt environmentally friendly behaviors such as reducing food waste. Food motivation and food preferences are linked to the moral and health elements of eating and influence the decision to purchase food accordingly [37,38]. Environmental concern is a key motivator for consumers to change their consumption habits and become more ecologically friendly [29]. Past studies have shown that environmental concern influences people's attitudes towards consuming green food positively [39]. Therefore, we establish the following hypothesis.

**H3.** *Environmental concern has a positive impact on the attitude towards reducing food waste.*

### 2.5. Marketing Addiction

Food waste behaviors of consumers are related to their attitudes, values, knowledge, and behavior towards food, lifestyle planning, and purchasing habits, as well as both their general consumption behaviors and their recycling or environmentally friendly behaviors [40]. Encouraging consumers by marketing incentives to believe that creating food waste by throwing away a large amount of food is a bad behavior may develop awareness and minimize the amount of wasted food. Promotions of the products and appealing packaging are examples of marketing methods which can encourage impulsive food purchases, which are one of the key factors of wasted food [41]. According to Mondéjar-Jiménez et al. [35], marketing methods may have an influence on consumer behavior regarding food waste. In addition, emphasis should be placed on fostering a positive attitude towards food waste and warning individuals about the negative effects of buying large quantities of food stocks, including over-purchases [42]. It was found that marketing addiction has a significant effect on perceived behavioral control [28]. Hence, we create the following hypothesis.

**H4.** *Marketing addiction negatively impacts perceived behavioral control towards the reduction of food waste.*

### 2.6. Subjective Norm

The intention is influenced by how the person perceives the pressure to partake in the behavior. The individual's impressions of what others anticipate them to do are known

as subjective norms. Certain conducts that are impacted by social influences, like their communities, friends or family, is referred to as a subjective norm. It can alter someone's behavior and performance [43]. This study therefore proposes that the intention to reduce the waste of food is influenced by the subjective norm. The subjective norm is one of the key elements of the TPB. Thus, it is included in this study in order to partially explain the intention to minimize wasted food. Additionally, a number of research demonstrated a relationship between the subjective norm and attitude, perceived behavioral control, and intention [44]. The subjective norm has a positive impact on perceived behavioral control and attitude in the extended TPB [27,45]. Additional studies show a positive effect of the subjective norm on attitude [31,46,47]. Thus, we created the following hypotheses.

**H5.** *The subjective norm positively affects perceived behavioral control towards reducing wasted food.*

**H6.** *The subjective norm positively influences the attitude towards reducing wasted food.*

**H8.** *The subjective norm positively affects the intention to minimize wasted food.*

### 2.7. Perceived Behavioral Control (PBC)

The individual's perceptions of their ability to partake in the behavior also influences their intention. Perceived behavioral control (PBC) refers to the people's perceptions of their ability to engage in the behavior, taking into account factors such as resources and past experiences [43]. According to Ajzen [43], intention is influenced by PBC. According to numerous studies, perceived behavioral control is a crucial part of intention [48]. As a result, the perception of the consumer's behavioral control factor has a direct impact on their intention. PBC relates to a person's assessment of the difficulty or simplicity of carrying out a particular behavior [43]. The purpose connected to a particular behavior, such as decreasing food waste, is influenced by perceived behavioral control [20]. PBC is one of the key elements of the TPB and may explain the intention to minimize wasted food. Thus, the following hypotheses are derived.

**H7.** *PBC positively affects the attitude towards reducing wasted food.*

**H9.** *PBC positively affects the intention to decrease wasted food.*

### 2.8. Attitude

The individual's attitude towards the behavior is one of the most important influences of intention. Attitudes are the individual's evaluations of the behavior and its outcomes [43]. One's views regarding conduct are said to be influenced by one's knowledge of the activity and its effects. For instance, the aim to reduce waste has been impacted by the awareness of the environmental effects of trash. Also, less food is wasted when people are aware of the issue [49]. One of the most efficient approaches to encourage sustainable food waste behavior is to raise awareness and understanding of the effects of wasted food on the ecosystem. The way a person feels about a certain activity determines whether people consider it to be positive or negative [43]. According to Graham-Rowe et al. [20], attitude may be one of the key factors influencing behavioral intentions such as food waste. Knowledge and attitudes towards food waste reduction have also been detected to influence the intention to minimize food waste, with those who are more aware and have positive attitudes being more likely to have the intention to reduce food waste [50]. Based on an existing paper, intention and attitude are positively correlated [51]. Attitude towards the environment is one of the key elements of the TPB and may explain the intention to minimize wasted food. Hence, we can derive the following hypothesis.

**H10.** *Environmental attitude positively impacts the intention to minimize wasted food.*

### 2.9. Intention to Reduce Food Waste

The TPB suggests that intention is the most accurate predictor of behavior. Three constructs—attitude, subjective norm, and perceived behavioral control—are used in the TPB to determine the intention [43]. The previously mentioned variables might have an influence on the intention to reduce food waste for consumers. According to Ahmed et al. [52], attitude, subjective norms, and perceived behavioral control have a positive impact on the consumers' intention to buy organic food. One of the key factors influencing behavioral intention, such as food waste, is attitude [53]. One study found that individuals who have a larger intention to decrease wasted food are more likely to participate in activities such as meal planning and storage [54]. Furthermore, the influence of personal and contextual factors, such as habits, planning habits and food surplus, have also been studied and found to impact the intention to reduce the amount of wasted food [34]. The United Nations indicates the reduction of food loss and waste as one of the environmental goals and includes the minimizing of food waste in the 2030 Agenda for Sustainable Development. In particular, Target 12.3 aims for halving global per capita food waste at the retail and consumer levels. This paper includes the intention as it is one of the key elements of the TPB. Moreover, the intention to minimize food waste and its influences is the major objective of this study.

### 2.10. Waste Preventing Behavior

Individuals who are more committed to preventing food waste tend to throw away less food [55]. Planning routines like checking inventories or preparing meals in advance, for instance, can help to cut the quantity of food wasted, whereas overcooking can increase food waste [56]. The avoidance of household food waste starts with buying behavior, where consumers are often swayed by various incentives including special deals and various psychological pitfalls [28]. To reduce food waste, helpful practices include pre-shopping planning and using shopping lists. Heidari et al. [28] concluded that waste preventing behaviors has an impact on food waste. Abdelradi [57] and Diaz-Ruiz et al. [40] provided evidence that waste-preventing behavior has a direct impact on the quantity of food waste. Also, regularly planning meals may help people estimate how much food they would need to buy and how much will be required to prepare those meals. Thus, we develop the following hypothesis.

**H11.** *Waste-preventing behavior positively impacts the intention to minimize wasted food.*

This study developed eleven hypotheses based on the literature review and suggests the following conceptual framework. The model examines the interactions of factors such as perceived ascription of responsibility, moral norm, environmental concern, marketing addiction, subjective norm, perceived behavioral control, attitude, intention to minimize food waste, and waste-preventing behavior that relates to the behavior of the food waste management by the population in Thailand. A black line indicates the impact of one factor on another factor.

## 3. Research Methodology

### 3.1. Pilot Study

In order to conduct the pilot study, we gathered consumer data from Thai consumers. We collected the data from 50 respondents in the Khon Kaen Province. This study's structure was examined using a 7-point Likert scale. In order to assess the validity of this study, the data from 50 respondents were utilized to examine the demographic coverage, common method variance (CMV), and Cronbach's alpha. Furthermore, the exploratory factor analysis (factor loadings) was applied to confirm the components used. The Cronbach's alpha measurement scale of the pilot test was acceptable. The exploratory factor analysis confirmed the factors and the number of items (Table 1). After that, this study used the results of the pilot test to change any unclear words and correct grammatical errors. Finally, the questionnaires for this study were prepared for distribution. The constructs, along with

their statements, are listed in the following table. The statements were scored and coded on a 7-point scale individually.

**Table 1.** Construct and measurement statements.

| Construct | Measurement Statement | Reference |
|---|---|---|
| Perceived Ascription of Responsibility | - My individual choices affect the environment either negatively or favorably.<br>- Each individual's contribution to waste reduction can help lessen the worldwide effect caused by food waste.<br>- By lowering household rubbish production, I can stop climate change. | [57] |
| Moral Norm | - I will produce less garbage if I am aware of how waste affects the environment.<br>- I will make less waste if I am aware that I produce more rubbish than residents of other towns.<br>- If I'm conscious of the fact that some of the food I throw away can feed some hungry people, I'll waste less. | [58] |
| Environmental Concern | - I feel responsible for the future generations due to environmental changes.<br>- I feel responsible for the environment.<br>- I can make efforts to protect the environment. | [58] |
| Marketing Addiction | - The layout and form of the goods in the supermarket compel me to purchase irrational items.<br>- Food packaging forces me to make unneeded purchases.<br>- I bought more than I needed to because of the supermarkets' specialized deals. | [58] |
| Subjective Norm | - The individuals who matter to me anticipate that I will practice environmental responsibility.<br>- People who matter to me advise me to think about protection of the environment initiatives.<br>- Relatives, friends, and community are anticipating to strive towards decreasing food waste in their homes. | [57] |
| Perceived Behavioral Control | - I have the option of wasting less food.<br>- I am struggling to prevent food waste in my house.<br>- I am fully responsible for wasting less food.<br>- I waste smaller amounts of food whether or not there are incentives in the neighborhood. | [58] |
| Attitude | - I would rather waste fewer food at home.<br>- Food is valuable in my perspective, thus wasting it is undesirable.<br>- The decrease in food waste in households is advantageous to me.<br>- It is important to inspire the decrease of wasting food. | [59] |
| Intention to Reduce Food Waste | - I intend to pay closer attention when I shop in the upcoming weeks to cut down on wasting food.<br>- I want to waste less food during the coming weeks by paying more attention to my eating.<br>- I intend to learn more in the coming weeks about the consequences of wasting food on the ecosystem and socioeconomic situations in my neighborhood. | [58] |
| Waste Preventing Behavior | - When I went shopping, I didn't use a plastic bag; I took my own bag.<br>- I strive to avoid making unnecessary purchases.<br>- I try to reuse things whenever I can.<br>- I prefer to purchase used goods than disposable ones. | [40,60] |

### 3.2. Sampling, Development of Measures, and Collection of Data

Quota sampling was used in this study's data collecting. In the four regions of Thailand during September and October 2022, 400 respondents provided the information, i.e., 100 respondents per region. The four main regions of Thailand are north, northeastern, central, and south. Face-to-face interviews with the help of a questionnaire were used to gather the data. A total of 369 replies were usable once irrelevant information, outliers, and errors were eliminated. Hence, 7.75% of the samples were deemed invalid. Participants were advised of confidentiality and research ethics for the business and social sciences before filling out the questionnaire. The respondents' information is kept confidential and the researcher did not share the data. The risk of a social desirability bias was minimized by ensuring the anonymity of responses and emphasizing the importance of honest answers. The two-part questionnaire was developed based on various studies. The first section of the questionnaire shows the social and demographic aspects of the respondents such as age, sex, income, and occupation (see Table 2). The second section shows the behavior

towards food waste [35,57,61,62]. Based on the paper of Ajzen [43], each structure contains multiple items. Participants were asked to rate their agreement with each statement on a 7-point Likert scale for each item (1 strongly disagree to 7 strongly agree). The sociodemographic data of the sample is shown in Section 4.1. The representativeness of the sample has been verified. The sample's characteristics (gender, income, etc.) can adequately reflect the population, as shown in Section 4.1. Further, random sampling within each of the four regions has been used to select the participants from the target population in order to ensure that each member of the population has an equal chance of being included in the sample. Moreover, the large sample size of 400 respondents lead to a more representative sample.

**Table 2.** Descriptive statistics.

| Demographic Variable | Categories | Total | |
|---|---|---|---|
| | | *n* | % |
| Segment size | | 369 | 100.0 |
| Gender | Female | 223 | 60.4 |
| | Male | 146 | 39.6 |
| Age | <21 | 12 | 3.3 |
| | 22–25 | 193 | 52.3 |
| | 26–35 | 77 | 20.9 |
| | 36–45 | 70 | 19.0 |
| | >45 | 17 | 4.5 |
| Occupation | Government service/Statement enterprise | 60 | 16.3 |
| | Private company employees | 24 | 6.5 |
| | General employees | 35 | 9.5 |
| | Freelance/Trading | 30 | 8.1 |
| | Students | 157 | 42.5 |
| | Business owner | 59 | 16.0 |
| | Other | 4 | 1.1 |
| Income | No income | 60 | 16.3 |
| | <20,000 | 135 | 36.6 |
| | 20,000–30,000 | 97 | 26.3 |
| | 30,001–40,000 | 31 | 8.4 |
| | >40,000 | 46 | 12.4 |

*3.3. Data Analysis and Statistical Measures*

The collected data were statistically examined by using SPSS 28 (IBM Corp., Chicago, IL, USA) and AMOS 28 (IBM Corp., Chicago, IL, USA). We investigated the common method variance (CMV) before examining the data. The results may include systematic error variances between the constructs and may have biased the studied relations. We used Harman's single-factor test [63]. The results showed a cumulative variance of 48.65% (smaller than the 50% limit), which confirmed the lack of CMV. The structural equation modeling (SEM) technique was utilized to analyze the study's data. The model's estimation was estimated using the SEM in two steps. The first step validates the model by determining the validity and reliability of each indicator's link to its variable. The goodness of fit (GOF), convergent validity (CV), and discriminant validity (DV) must all be evaluated in this step. The specified thresholds for the GOF and convergent validity conditions were CMIN/df < 3.00, RMSEA < 0.10, CFI > 0.90, CR > 0.70, and AVE > 0.50. Regarding the condition of discriminant validity, this paper looked at multicollinearity problems and the identity matrix of the constructs. To test for multicollinearity, the paper applied Pearson's moment correlations with a cutoff of 0.80. Using Bartlett's sphericity and the Kaiser–Mayer–Olkin (KMO) tests, an identity matrix was evaluated. These requirements were met. The second step assesses the structural equation model to see if the whole model—including the GOF estimation—is reliable. CMIN/df < 3.00, RMSEA < 0.10, and CFI > 0.90 were set to fit the indices' thresholds.

## 4. Result of the Study

### 4.1. The Sample

According to Table 2, the sample size was 369 people. A total of 60.4% of those who responded were female, while 39.6% were male. In terms of age, the bulk of respondents were 22–25 years old, accounting for 52.3%, followed by 26–35 years old (20.9%), 36–45 years old (19.0%), and the elderly (4.5%). A total of 3.3% of the responders were under the age of 18. The majority of responders (42.5%) were students, followed by government employees, state companies, and business owners (16.0%). In terms of the other occupations, freelance/trading was 8.1%, general employees were 9.5%, private employees were 6.5% and the others were 1.1%. It is evident that the majority of the respondents (36.6%) earn less than 20,000 THB per month. There are 146 men and 223 females in the sample. The majority of the participants were between the ages of 22 and 25, students, and earned less than 20,000 baht each month.

### 4.2. Measurement Model

#### 4.2.1. The Goodness of Fit (GOF)

Table 3 displays the GOF measurements and its associated limits. The outcomes met our expectations. The measurements all met the necessary standards. CMIN/df (1.989), the Tucker–Lewis index (TLI; 0.945), the incremental fit index (IFI; 0.990), the comparative fit index (CFI; 0.988), and the root mean square error of approximation (RMSEA; 0.084) passed the specified criteria.

**Table 3.** The GOF of the Measure Model.

| Indicator | Number | Criteria | Result |
|---|---|---|---|
| *p*-value | 0.000 | | Acceptable |
| CMIN/df | 1.989 | <3.00 | Passed |
| TLI | 0.945 | >0.90 | Passed |
| CFI | 0.988 | >0.90 | Passed |
| IFI | 0.990 | >0.90 | Passed |
| RMSEA | 0.084 | <0.10 | Passed |

#### 4.2.2. Convergent Validity (CV)

By comparison of the model's output with the index threshold, convergent validity was determined. The average variance extracted (AVE), composite reliability (CR) and Cronbach's Alpha ($\alpha$) were computed. The calculated indicators and suggested criteria for the CV measures are presented in Table 4. Furthermore, Table 4 displays the PAR (Perceived Ascription of Responsibility), MN (Moral Norm), EC (Environmental Concern), MA (Marketing Addiction), SN (Subjective Norm), PBC (Perceived Behavioral Control), AT (Attitude), IRFW (Intention to Reduce Food Waste), and WPB (Waste Preventing Behavior), all of which passed the CV requirements when the estimated outcomes were compared to the thresholds.

**Table 4.** Convergent validity.

| Construct | Indicator | *p*-Value | AVE | CR | $\alpha$ |
|---|---|---|---|---|---|
| Perceived Ascription of Responsibility (PAR) | PAR1 to 3 | *** | 0.723 | 0.871 | 0.825 |
| Moral Norm (MN) | MN1 to 3 | *** | 0.693 | 0.833 | 0.795 |
| Environmental Concern (EC) | EC1 to 3 | *** | 0.715 | 0.834 | 0.799 |
| Marketing Addiction (MA) | MA1 to 3 | *** | 0.631 | 0.787 | 0.754 |
| Subjective Norm (SN) | SN1 to 3 | *** | 0.697 | 0.822 | 0.788 |
| Perceived Behavioral Control (PBC) | PBC1 to 4 | *** | 0.525 | 0.716 | 0.703 |
| Attitude (AT) | AT1 to 4 | *** | 0.737 | 0.841 | 0.812 |
| Intention to Reduce Food Waste (IRFW) | IRFW1 to 3 | *** | 0.632 | 0.719 | 0.708 |
| Waste Preventing Behavior (WPB) | WPB1 to 4 | *** | 0.570 | 0.881 | 0.850 |

Note: *** Significant at <0.01.

### 4.2.3. Discriminant Validity (DV)

The degree of difference between two or more theoretically related constructs is known as DV. This was determined by contrasting the correlations of the relevant matrices with the square root AVEs. As indicated in Table 5, the square root of each AVE was greater than the off-diagonal correlation coefficients, suggesting that all constructs may conceivably be different constructions. The results of the DV were satisfactory.

**Table 5.** Discriminant validity.

| Construct | PAR | MN | EC | MA | SN | PBC | AT | IRFW | WPB |
|---|---|---|---|---|---|---|---|---|---|
| PAR | 0.850 | | | | | | | | |
| MN | 0.485 | 0.832 | | | | | | | |
| EC | 0.683 | 0.800 | 0.846 | | | | | | |
| MA | 0.413 | 0.407 | 0.452 | 0.794 | | | | | |
| SN | 0.502 | 0.492 | 0.503 | 0.280 | 0.835 | | | | |
| PBC | 0.394 | 0.600 | 0.562 | 0.398 | 0.519 | 0.725 | | | |
| AT | 0.628 | 0.737 | 0.808 | 0.361 | 0.569 | 0.610 | 0.858 | | |
| IRFW | 0.503 | 0.559 | 0.672 | 0.447 | 0.380 | 0.556 | 0.749 | 0.795 | |
| WPB | 0.571 | 0.721 | 0.802 | 0.386 | 0.457 | 0.660 | 0.834 | 0.750 | 0.755 |

### 4.3. Primary Structural Model

We integrated the constructs to create the structural model, as shown in Figure 1. In addition, we investigated the variables using the primary goal structure model. Most goodness of fit (GOF) criterion findings indicate how the structures support one another. The thresholds of all GOF indexes were met (see Table 6).

**Table 6.** The GOF of the Structural Equation Model.

| Indicator | Number | Criteria | Result |
|---|---|---|---|
| *p*-value | 0.000 | | Acceptable |
| CMIN/df | 1.982 | <3.00 | Passed |
| TLI | 0.944 | >0.90 | Passed |
| CFI | 0.985 | >0.90 | Passed |
| IFI | 0.992 | >0.90 | Passed |
| RMSEA | 0.087 | <0.10 | passed |

The results of the structural equation model give important insights into the relationships between the various constructs related to food waste behavior. The test supported several hypotheses, indicating a significant influence of various factors on behavior and attitude towards reducing food waste. Figure 2 and Table 7 show the findings of the SEM. The blue arrows represent hypotheses which are not supported. H1, H3 to H6, and H9 to H11 are supported at a significant level of 0.01 or lower, while H2, H7, and H8 were not supported. The following constructs were taken into account by the authors as they built the analysis: Perceived Ascription of Responsibility (PAR), Moral Norm (MN), Environmental Concern (EC), Marketing Addiction (MA), Subjective Norm (SN), Perceived Behavioral Control (PBC), Attitude (AT), Intention to Reduce Food Waste (IRFW), and Waste Preventing Behavior (WPB) to the theory of planned behavior. The first hypothesis was confirmed, showing that the perceived ascription of responsibility has a positive effect on the subjective norm. The standardized factor loading is 0.943. The second hypothesis was rejected, i.e., the moral norm has no impact on attitude. The third hypothesis was supported, indicating that environmental concern positively affects peoples' attitude. The factor loading is 0.779. The fourth hypothesis predicted that the impact of marketing addiction on PBC is negative to food waste management; it was also supported with a standardized estimate of −0.109. The fifth hypothesis was supported, suggesting that the subjective norm positively affects PBC to food waste management. The estimate is 0.911.

Moreover, the sixth hypothesis was confirmed and, therefore, the subjective norm positively impacts peoples' attitude towards food waste management (factor loading = 0.635). The seventh hypothesis was disproved, i.e., perceived behavioral control has no effect on attitude. The eighth hypothesis was also rejected, i.e., the subjective norm has no impact on the intention to minimize wasted food. The ninth hypothesis was supported, implying that perceived behavioral control positively affects the desire to minimize wasted food. The standardized factor loading is 0.712. The tenth hypothesis was supported, i.e., attitude has a positive impact on the intention to reduce wasted food (standardized estimate = 0.625). Finally, the eleventh hypothesis was supported, i.e., waste preventing behavior positively impacts the intention to minimize wasted food with a standardized factor loading of 0.843. These results support the importance of considering the various factors when addressing food waste and behavior towards reducing it. The most important variable in determining the intention to reduce food waste is waste preventing behavior which is highly significant.

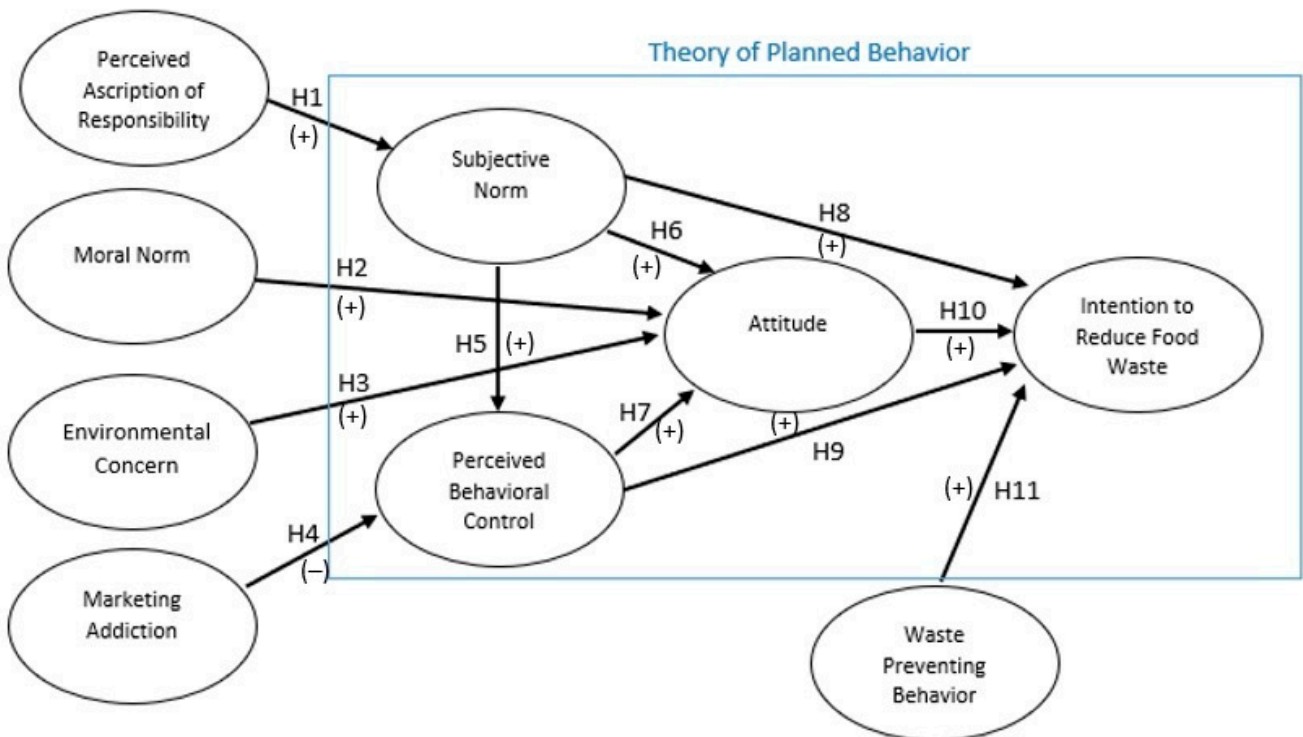

**Figure 1.** Proposed Model. Source: Illustration created by the authors (2023).

**Table 7.** Findings of the structural equation model.

| Hypothesis | Relationship | Standardized Estimate | t Value | Finding |
|---|---|---|---|---|
| H1 | PAR → SN | 0.943 *** | 2.856 | Supported |
| H2 | MN → AT | −0.104 | 1.114 | Rejected |
| H3 | EC → AT | 0.779 *** | 4.131 | Supported |
| H4 | MA → PBC | −0.109 *** | 3.146 | Supported |
| H5 | SN → PBC | 0.911 *** | 2.955 | Supported |
| H6 | SN → AT | 0.635 *** | 6.860 | Supported |
| H7 | PBC → AT | −0.743 | 0.886 | Rejected |
| H8 | SN → IRFW | −0.914 | 0.644 | Rejected |
| H9 | PBC → IRFW | 0.712 *** | 5.579 | Supported |
| H10 | AT → IRFW | 0.625 *** | 3.397 | Supported |
| H11 | WPB → IRFW | 0.843 *** | 7.511 | Supported |

Note: *** Significant at <0.01.

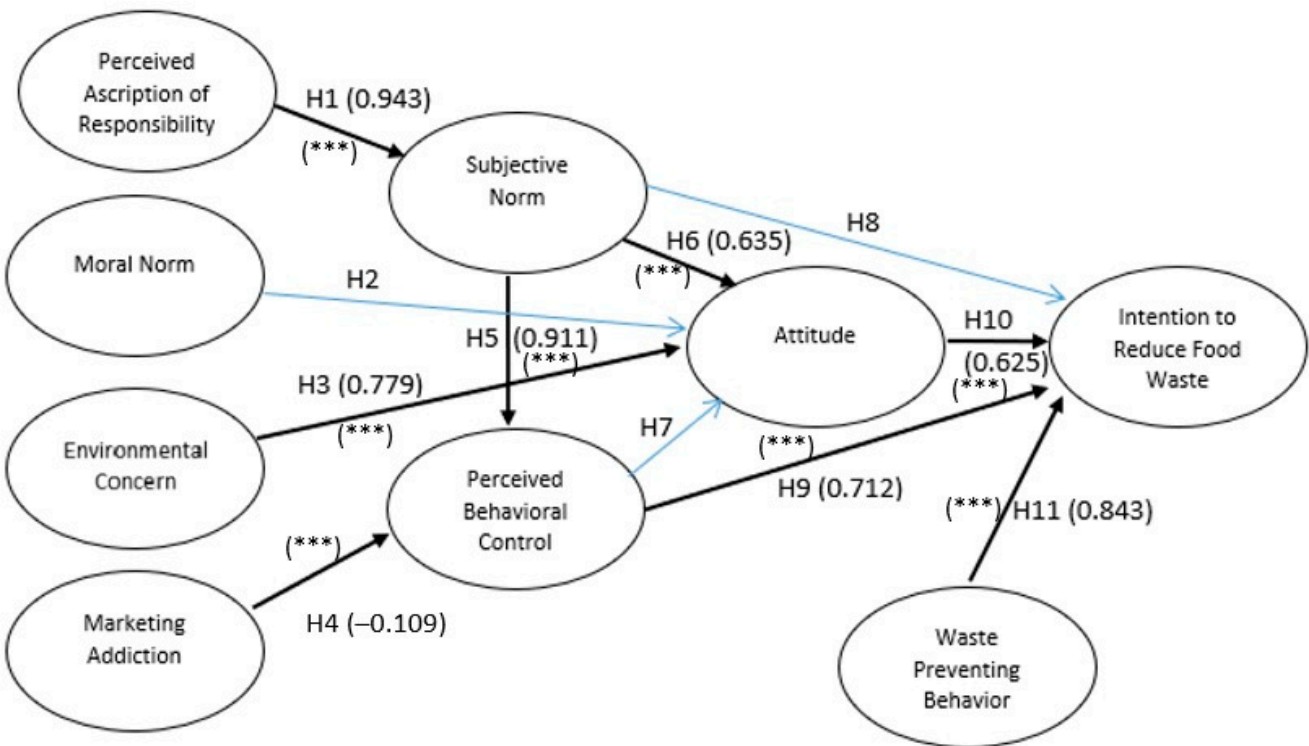

**Figure 2.** The SEM with results. Source: Illustration produced by the authors (2023). Note: *** Significant at <0.01; black (blue) arrows indicate a significant (insignificant) relationship.

## 5. Discussion

### 5.1. Research Implications

By examining several variables, the purpose of this study is to thoroughly explain the desire to reduce food waste. Consumers who have a perceived ascription of responsibility, environmental concerns, and less marketing addiction towards wasted food have a larger aim to minimize food waste, according to this study. This impact is formed through the effect on perceived behavioral control, the subjective norm, and attitude, which is consistent with a recent study by Razali et al. [64]. Furthermore, consumers with waste preventing behavior show a larger desire to minimize food waste. Mondéjar-Jiménez et al. [35] discovered similar results in the food waste behavior of young individuals. Perceived behavioral control has a large effect on the desire to decrease food waste and is a key item that can clarify the intention to minimize the amount of food waste. This finding is consistent with a recent article by Coskun and Özbük [33]. Attitude has a positive effect on the intention to minimize wasted food, which is coherent with a recent paper by van der Werf et al. [65] and a study by Kim et al. [66]. Prior research has shown that the subjective norm is an essential element in describing the desire to minimize food waste [35,67]. This study finds no significant relationship between the subjective norm and the intention to decrease wasted food. People who are influenced by what others think about food waste reduction may not necessarily impact one's intentions to minimize wasted food. However, a significant factor of the intention to decrease food waste was found to be waste preventing behavior. Individuals who engage in behaviors to prevent food waste are more likely to have the intention to reduce the waste of food. Researchers can use this finding to develop interventions and educational programs that specifically target waste preventing behaviors. By promoting and encouraging behaviors such as proper food storage, meal planning, portion control, and utilizing leftovers, individuals can actively participate in reducing food waste. Consumers who practice waste reuse, the reduction of waste, and recycling create less food waste. Policymakers can use this research finding to inform the development of policies and initiatives aimed at promoting waste prevention

behavior. This can include implementing regulations or guidelines related to food storage, expiration date labeling, portion sizes, or food donation practices. Additionally, incentives or rewards can be introduced to encourage households, supermarkets, and restaurants to prioritize waste prevention. The moral norm has no impact on attitude, implying that the moral values and beliefs of individuals do not significantly impact their attitudes towards reducing food waste. Further, environmental concerns have a positive effect on attitude, which subsequently impacts the intention to minimize wasted food. Consumers with a high degree of environmental concern may reconsider their attitude towards wasted food and may decrease their waste of food. Marketing methods and addiction have a negative effect on perceived behavioral control. This finding is consistent with the existing literature [28]. Individuals who are addicted to marketing are less likely to perceive that they have control over reducing food waste. The buying environment may enhance food waste creation and may have an impact on consumer behavior. Marketing methods (marketing addiction) and appealing packaging have a negative effect on perceived behavioral control, and hence, lead to a rise in food waste. As a result, governments may adopt food waste reduction initiatives and may also influence the marketing methods and packaging of food. These initiatives can influence the behavior of consumers and lead to an increased waste preventing behavior. Food waste reduction programs need to concentrate on TPB constructs, for example, the subjective norm, perceived behavioral control, and attitude [53]. This paper's findings support this. Moreover, perceived ascription of responsibility positively impacts the subjective norm, which in turn affects attitude. Individuals who perceive that they have a higher degree of responsibility for reducing food waste are also more likely to be influenced by the opinions and expectations of others (subjective norm) regarding reducing food waste. Thus, organizations and governments may strengthen the responsibilities of individual consumers by raising awareness for waste preventing behavior, which leads to an increased overall responsibility for food waste. Programs on reducing food waste and the effects of food waste on the environment may be developed. Through these activities, people can learn about the impacts of wasted food and be inspired to prevent it. Therefore, the environmental concern of consumers can be increased, which subsequently affects the attitude towards food waste. Techniques that support the subjective norm and the attitude that wasting food is wrong should be created. The attitude, perceived behavioral control, and waste preventing behavior are the most important factors which can directly affect the reduction in wasted food. Interventions and policies aimed at reducing food waste should target these factors. For instance, campaigns and educational programs that aim to change individuals' attitudes towards food waste could be effective in promoting waste-reducing behaviors. Additionally, programs that aim to increase individuals' perceived behavioral control, such as providing information on food waste, could also be effective. Policies and regulations that address waste management practices could also be implemented to support waste-reducing behavior. The results provide valuable information about the impact of various factors on the behavior and attitude towards reducing food waste. This can help researchers and policymakers to understand which factors are crucial in shaping individuals' behaviors and attitudes towards reducing food waste. By understanding the factors that impact the behavior and attitude, policymakers can design policies and initiatives that inspire consumers to reduce the quantity of wasted food and promote sustainable practices.

### 5.2. Research Limitations & Future Research

This paper enhanced the original TPB model satisfactorily to describe food waste behavior. The results provide a foundation for further research on the relationships between the various constructs and food waste behavior. There are still certain limitations on this study. To further understand customers' behavioral intentions, future studies may incorporate additional variables, such as social and financial situations, into the existing structural equation model. Future studies may include more factors (such as food prices, family background, and financial factors) in the extended TPB model. These initiatives would aid in improving the comprehension of the desire and actions to decrease food waste.

Moreover, this research is based on a sample from Thailand. Future research may focus on other Asian countries in order to extend the sample. Additionally, because of cultural, awareness, educational, infrastructure, and policy differences, the study may be conducted in European countries, which may provide different results. The limited sample size of 400 respondents may limit the generalizability of the findings to larger populations.

## 6. Conclusions

Nowadays, consumers have clearly given importance to food waste as it has a damaging effect on the world and the environment. We modeled the intention to minimize wasted food in Thailand. Hence, the theory of planned behavior's main variables and new concepts were combined in a structural equation model. This article focuses on the significant relationships between several factors which relate to consumer behavior regarding the reduction of the quantity of food waste. We executed quantitative research using questionnaires with 369 valid respondents who were aware of and focused on the issue of wasted food in Thailand. The results suggest that Perceived Ascription of Responsibility, Environmental Concern, Marketing Addiction, Subjective Norm, Attitude, Perceived Behavioral Control, and Waste Preventing Behavior all play a role in shaping individuals' intentions to minimize wasted food. The attitude, perceived behavioral control, and waste preventing behavior are the main predictors of the intention to decrease the amount of wasted food. This finding guides the development of behavior change strategies, educational programs, policies, and future research efforts to encourage waste prevention and ultimately reduce food waste. Consumers who engage in behaviors to prevent food waste are more likely to have the intention to minimize wasted food. Further initiation and support of attitude, waste preventing behavior, perceived behavioral control, and environmental concern may help to increase consumers' intention to minimize wasted food, such as developing educational campaigns highlighting the impacts of food waste and providing practical tips for waste reduction, using signage and labels to encourage portion control, providing recipe ideas for using leftovers, improving access to food storage guidelines, composting facilities, and meal-planning apps. The results provide useful insights into the relations between several constructs and food waste behaviors, and they may help to form interventions aimed at decreasing food waste.

**Author Contributions:** Conceptualization, J.S.; methodology, J.S.; formal analysis, A.H., S.T., N.S. and K.C.; resources, J.S.; data curation, A.H., S.T., N.S. and K.C.; original draft preparation, J.S.; review and editing, C.K.; supervision, J.S.; project administration, C.K. All authors have read and agreed to the published version of the manuscript.

**Funding:** The Khon Kaen University International College Research Grant (contract 40/2564 by KKUIC) provided funding for this research.

**Institutional Review Board Statement:** Not applicable.

**Informed Consent Statement:** Informed consent was obtained from all subjects involved in this study.

**Data Availability Statement:** The data are contained within the article.

**Acknowledgments:** All authors thank the International College, Khon Kaen University, Thailand and the Center for Sustainable Innovation and Society for offering the use of their research facilities.

**Conflicts of Interest:** The authors declare no conflict of interest.

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
