# Peer review of "Factors of Food Waste Reduction Underlying the Extended Theory of Planned Behavior: A Study of Consumer Behavior towards the Intention to Reduce Food Waste"

_resources, doi:10.3390/resources12080093_

Round 1

Reviewer 1 Report (Previous Reviewer 3)

This paper is much improved and better supported. There remain some issues with English language and, as such, editing to ensure accuracy of language is suggested.

As above - some issues with English expression, although these do not obscure meaning

Author Response

Please see this attachment. Thank you.

Reviewer 2 Report (New Reviewer)

In general, the literature review is not presenting the UN environmental goals and should be added to 

this article.

Page 1 Line 29

Put keywords in alphabetical order.

Page 7 Line 273

Can´t find any information or section discussing Bias in this article.

Have someone to proof read the English. 

Author Response

Reviewer 3 Report (New Reviewer)

The research presented is interesting, however there are a few issues to be considered.

First, up-to-date reference to be provided. There are many studies performed in the last  5-6 years which were not mentioned. Moreover, teh statement  "Around one-third of the food produced worldwide is never consumed" although true it should be supported by a  newer reference (2012 is rather old for such statistics) 

Second, the aim and specific objectives should be stated at the end of the Introduction section and also in the abstract. One can find this in the paper but it is not consistent all over the paper. For example, in the line 324 it says that the aim of the article is to encourage people to learn about reducing food waste - don't really agree since scientific articles are read mainly by scientists. 

Third, the gap in the literature is not very well defined. 

The discussion section should also be revised, by referring to more up-to-date references.

Author Response

Please see the attached file. Thank you.

Reviewer 4 Report (New Reviewer)

I appreciate the authors' attitude to the treatment of the subject. The presented paper is processed out at a high level and it demonstrates the authors' orientation in it. In spite of the above mentioned, I have a few comments:

- I miss the verification of the representativeness of the sample,

- the abstract states that the research was targeted both on households and individual consumers, but the research rather implies that it was conducted on individual consumers - it would be useful to clarify this,

- there are several parts in the paper marked in different colors - either this is the authors' working version or an unrecognized mistake,

- the term percentage is also used in the text, but it would be more appropriate to refer to it uniformly by the % symbol,

- some parts of the article do not fully copy the article template - sections 275-284 need to be aligned and references should be cited according to the template,

- after subsection 2.1 Theory of Planned Behavior there is a blank half page - it would be more appropriate if it was immediately followed by subsection 2.2 Perceived Ascription of Responsibility,

- Table 1 should be relocated under subsection 3.1 Pilot study.

Round 2

Reviewer 3 Report (New Reviewer)

The manuscript was improved based on recommendations.

This manuscript is a resubmission of an earlier submission. The following is a list of the peer review reports and author responses from that submission.

Round 1

Author Response

Dear Reviewer 1,

We appreciate your feedbacks. Please see the attached file.

Best,

Reviewer 2 Report

The manuscript analyses factors affecting the intention to minimize food waste using an extended Theory of Planned Behaviour.

The entire manuscript is written with numerous repetitions. Several sentences were written twice in a row with different syntax and synonyms. As a result, the entire manuscript becomes unnecessary long.

The introduction sufficient to provide an overview about the topic and the research question. Nevertheless, it lacks a clear structure which causes a lot of repetitions.  Beside that, some statements are a kind of generic and should be based on statistics and references (e.g., 39-40: “People can cause food wase without realizing that it affects other people and therefore, most people ignore it”, 43-44: “Thailand is one of the countries in the world were food waste is a significant problem”).

The section pointing out the environmental problems caused by food waste (44-50), is ignoring that the main problem of food waste is that it is connected with a unnecessary high agricultural production. It is not clear if the authors refer to that implicit or if the food waste causes the environmental burdens by it existence itself (e.g. treatment of waste).

Although it is correctly pointed out that private households are a relevant factor at the level of consumption, no relation is established to upstream stages of the value chain.

Illustrate methods to reduce food waste, encourage people to learn about food waste and develop policies to do so stop environmental problems (97-100) was formulated as the aim of the study. This is not correct and is not accomplished by the study.

The section Literature Review presents the initial elements of the TPB in a sufficient way. In the description of the extended elements (section 2.2 – 2.6), the first part of each section is always a repetition of the introduction (food waste cause environmental problems and private households are responsible for food waste). In contrast, the development of the hypothesis is often neglected (esp. in case of H1 and H4). Section 2.6 does not shows sufficient that “Waste preventing behavior” means waste prevention in other consumption fields. This is first clearly shown by table 1.

The description of the Research Methodology is sufficient. The items used to measure “moral norm” are however questionable and should be revised. The first item addresses e.g. environmental concern but not own moral convictions.

The Results are presented sufficient.

The Discussion in particular is characterized by countless repetitions. Up to line 457, only the results are repeated and not discussed or compared with other results from the literature. Also in the following section, the discussion remains highly superficial and does not derive any new insights from the results.

The suggestion „Organizations might develop programs on protecting the environment and the environmental effects of wasted food” is not a new finding and has nothing to do with the results of the study. It is not concluded from the different elements of the extended TPB which measures could be helpful to avoid food waste and what the influence of the different elements implies.

The last section does not provide a conclusion but only summarizes the entire article once again.

Author Response

Dear Reviewer 2,

Thank you for your valuable feedbacks. Please see the attached file.

Best,

Reviewer 3 Report

Amend title - the English here is clumsy

Ensure all assertions through the paper are evidenced

It would good to talk more about the outcome of studies using TPB - the fact of using the theory does not indicate it was successful.

I do not think you need to justify the elements of the TPB - these are standard requirements of the model.

2.2 - Humans need to be aware -  this sentence is not needed

Distinguish edible and inedible food waste somewhere in the paper

2.4 - Food choices in the future - this sentence does not make sense

Methods - I doubt that you used random sampling - if you did you need to evidence this through more description of sampling approaches

Ensure you refer to tables from the text

Did you do a pre-test or pilot?

Fig 2 - are all the relationships shown significant? - I find it hard to believe H4 was significant

Discussion of results is very descriptive, it would benefit from more comparison with the literature

Limitations identified could be expanded

Conclusion is too long and includes unsubstantiated assertions

Did you consider social desirability bias in responses?

Did you measure Chronbach's alpha for constructs?

I am not convinced of the validity of some of the questions (subjective norm, PBC, intention)

Author Response

Dear Reviewer 3, 

We appreciate your comments. Please see the attached file.

Best,

Round 2

Reviewer 2 Report

Although many parts of the manuscript have been marked as revised, in many cases nothing has been changed in these parts and they are still in their original form (e.g. first part of discussion, whole conclusion). It should therefore be noted that some of the issues noted in the previous review have been revised, but many have not.

Overall, a comprehensive linguistic and structural revision of the article is recommended.

The introduction is still very unstructured which causes a lot of repetitions. In addition, the entire section on the objective of the study and the rationale has not been improved, but has been completely removed.

The section Literature Review was also not significantly improved. Still many statements of the introduction are just repeated and the deduction of the elements is very short or missing. E.g.,

-       In 2.2, the only sentence justifying the H1 has been removed. Instead, there are only statements that can already be found in the introduction. The section 2.10 is very confusing. It is not clear if waste prevention behavior refers to food waste or waste in other consumption fields. (242-245: “Waste preventing behavior refers to waste prevention in other consumption fields“ vs “Planning routines like checking inventories or preparing meals in advance, for instance, can help to cut the quantity  of food wasted, whereas overcooking can increase food waste“)

-       In 2.5 The main part of the section (160-167) does not explain the element marketing addiction.

-        

All in all, each section of the literature review should be revised.

The description of the Research Methodology is sufficient. The items used to measure “moral norm” are however questionable and should be revised. The first item addresses e.g. environmental concern but not own moral convictions.

The Results are presented sufficient.

The Discussion in particular is characterized by countless repetitions. Up to line 457, only the results are repeated and very limited discussed or compared with other results from the literature. Also in the following section, the discussion still remains highly superficial and does not derive any new insights from the results.

Also the suggestion or implications like„Organizations might develop programs on protecting the environment and the environmental effects of wasted food” are very generic and do not provide new findings based on the results of the study.

The last section still does not provide a conclusion but only summarizes the entire article once again. Although the entire section is marked in red, almost nothing has been modified in this section.

Therefore, I still do not recommend the publication of the manuscript in the present form.

Reviewer 3 Report

The authors have addressed most of my previous concerns.

The Hypothesis development section could still be better supported.

Line 177 - although the authors cite one (of many?) studies that have found a relationship between SN and PBC or Attitude, this is not what the TPB model expects. Hence, unless this is better supported I would remove H5, 6, 7.

You have indicated that there was a pilot, but this is poorly explained and we have no idea of the outcome of this and thus how it influenced the instrument used. I would also suggest that the use of EFA would help in confirming the factors used. The pilot should also come after the instrument development.

You have given references for the measures used, but I am not clear why these have been selected over alternatives. I also have the same concerns about some of the questions. For instance, the first question related to responsibility is poor - it comprises two questions in one, I do not think 2 out of 4 of the attitude items measure attitude, and 2 of 4 of the waste-related behaviours are not food related. If EFA loaded these items as you expect, the work would be more convincing.

Under limitations you refer to other Asian countries - what about beyond Asia and why would that be interesting?

Conclusions are still too long. They could start at Results in line 536.